# The nexus between environmental concern and future childbearing aspirations among university students in Bangladesh

**Bijoya Saha, Shah Md Atiqul Haq**[ORCID]*

Department of Sociology, Shahjalal University of Science and Technology, Sylhet, Bangladesh

* shahatiq1@yahoo.com, shahatiq-soc@sust.edu

## Abstract

This study investigates the relationship between environmental concerns and future childbearing aspirations among university students in Bangladesh. It included 380 final-year and master's students from various academic disciplines at Shahjalal University of Science and Technology (SUST), Bangladesh, who completed a structured questionnaire. The binary logistic regression model and Poisson regression model were employed to examine the effects of these variables (environmental concern, gender, religion, academic performance, university courses related to the environment or climate change, field of study, and perceived vulnerability to extreme weather events or climate change in their home region) on intentions to have children in the future. The findings demonstrate that environmental concerns significantly impact university students' intentions to have children. Additionally, students who are concerned about environmental issues are more likely to desire children in the future and plan to limit their family size due to these concerns. Students' future parenting plans are strongly influenced by their perceptions of environmental challenges. According to this study, female students are less likely to intend to have children if environmental conditions worsen. The findings suggest that several factors, including gender, disciplinary background, and environmental degradation, may influence future fertility intentions and, consequently, affect population dynamics. Such factors may also play a crucial role in shaping future population policies aimed at addressing the effects of climate change and achieving environmental sustainability.

## 1 Introduction

Environmental concerns can range from a specific attitude toward ecologically acceptable activities to a broader moral orientation [1]. Major environmental problems include resource depletion, pollution, climate change, and environmental degradation. Researchers now claim that the effects of climate change make social and behavioral studies on environmental issues more useful [2]. Studies indicate that various social

**Data availability statement:** All data are in the manuscript and/or supporting information files.

**Funding:** The author(s) received no specific funding for this work.

**Competing interests:** The authors declare that there is no conflict of interest.

structural elements such as gender, education, age, employment, and political orientation have a significant impact on environmental concerns [3]. Research from different countries indicates that environmental attitudes vary by demographic groups, with undergraduate women in Canada expressing greater concern due to emotional sensitivity, whereas men in Nigeria demonstrate higher awareness and knowledge of environmental issues [4]. Additionally, a survey of ten countries highlights the emotional and psychological impact of climate change on young people who express anxiety and dissatisfaction with government climate change policies [5].

Concerns about climate change influence reproductive decisions globally; surveys and media coverage indicate that young people may decide not to have children due to environmental degradation [6–10]. Some individuals think that childbearing exacerbates the ecosystem, while others fear that climate change could negatively impact children's health [8,9]. Climate change may indirectly affect fertility decisions [11], influencing reproductive behaviours through factors such as temperature and precipitation [12]. In some cases, climate change may prompt families to adjust their family size in response to resource scarcity. For example, Thiede argues that women might have more children in response to infant mortality linked to climate change [11]. Ghimire and Mohai discovered that having children is less likely in Nepal when there are environmental issues [13], while higher birth rates and the desire for larger families are linked to a greater reliance on natural resources that are held by poorer environmental quality [14]. In a study of Canadian university students, Arnocky et al. [1] found a relationship between environmental concerns and fertility intentions, with pollution concerns leading to lower fertility intentions.

Globally, densely populated nations face many challenges due to environmental degradation and resource scarcity, which could influence reproductive aspirations. Countries with high population densities often struggle with land degradation, extreme weather events (EWEs), and food security factors that may shape fertility preferences. For instance, in a nation such as Bangladesh, India, China, or Malaysia, rapid population growth and environmental vulnerabilities intensify the concerns about the sustainability of future generations [12,15–17].

## 1.1  Context of the study: Bangladesh

Bangladesh is one of the densely populated Asian countries [18] and is highly vulnerable to climate change due to its low-lying river delta, lengthy coastline, and floodplains that span 80% of the country [19]. Bangladesh ranks sixth in terms of vulnerability to climate change, having experienced 185 severe weather occurrences linked to climate change [20].

Despite these vulnerabilities, Bangladesh remains a resilient nation with natural resources and extensive river networks that are essential to its economy and support livelihoods [21]. However, climate change intensifies existing challenges, including drought, river erosion, cyclones, storm surges, hotter and drier summers, heavy rainfall, and disruptions to agricultural production and water security. These environmental pressures have a broad impact on the people of Bangladesh, affecting economic instability, migration decisions, and the reproductive decision-making process [22].

According to the Bangladesh Bureau of Statistics Report [23], approximately 91.04% of Bangladesh's total population identifies as Muslim, 7.95% as Hindu, 0.61% as Buddhist, 0.30% as Christian, and 0.1% as belonging to other religions. Around 48% of the total population is under the age of 25 [23]. Of this group, approximately 37% are enrolled in university-level education, with a majority being male, highlighting significant youth engagement in higher education [24]. According to the United States government, 90 million Bangladeshis, or 56% of the population, live in "high climatic exposure areas," with 53 million vulnerable to "extremely high" exposure [25]. Additionally, university students come from different geographic locations, and many of them are susceptible to environmental risk. These demographic patterns are crucial contexts for understanding fertility intention, as factors such as gender, education, and perceived climate vulnerability have been shown to influence future childbearing intentions in both regional and global studies.

This study examines how environmental issues specifically influence a student's decision-making regarding having children. Environmental concern in this study refers to students' apprehension about environmental degradation, such as pollution or climate change, which may impact future generations, particularly children's health and well-being. Future childbearing aspirations are defined as students' intention to have children in the future, ranging from plans to have multiple children to choosing to remain child-free. Given that university students in Bangladesh are increasingly aware of family planning [26] and concerned about environmental issues [27], it is crucial to investigate their reproductive intentions in relation to environmental concerns since many people in the country are under the age of 25. The study addresses key questions: How do university students in Bangladesh perceive the relationship between environmental concerns and their reproductive decisions? Is there a link between environmental concerns and intention to have children in the future in the context of Bangladesh? While the notion of environmental concerns is not new, the relationship between environmental concerns and future childbearing intentions is relatively recent. By knowing how university students view the impact of environmental issues on their intention to have children in the future, the country can better plan how to maximize the involvement of the next generation to achieve climate change resilience, women's empowerment, and environmental sustainability. In terms of predicting future population structure concerning climate change and environmental issues through future studies, the findings of this study reveal new dimensions and may provide undiscovered factors that influence students' reproductive decisions in the future. Examining the intriguing relationship between the sociodemographic factors of university students and their aspirations to have children in the context of environmental degradation, especially in developing countries, can provide insights into the motivations behind fertility preferences among the younger generation. Such understanding may have important implications for achieving national fertility targets.

## 2 Literature review

Researchers are increasingly focused on the relationship between environmental concerns and the intention to have children. A lot of studies have been undertaken on the subject, with varying degrees of results. Previous research has shown a complicated relationship between environmental challenges and reproductive decisions.

### 2.1 Factors associated with fertility decision-making

Research on fertility intention suggests that social and psychological factors such as personal fulfillment, societal expectations, gender, education, religious beliefs, and home areas influence reproductive decision-making [28]. For example, women are less likely than men to express robust negative intentions regarding having children [29]. Females are more likely to use contraceptives than males among the Indigenous people of Bangladesh [30]. In many developed countries, a higher education level among couples is associated with increased fertility aspirations [31,32]. Additionally, individuals with more education tend to have higher contraceptive usage rates [13]. Compared to humanities students, science students demonstrate a higher awareness of fertility and a more realistic understanding of fertility-related issues [33]. Religious beliefs play a crucial role in shaping reproductive decisions. For example, the Muslim community tends to have higher fertility rates due to religious encouragement for larger families, restrictions on contraception, and cultural norms on

childbearing as a social and cultural duty [34,35]. In terms of vulnerable areas, Ahmed and Haq [36] revealed that women living in climate-vulnerable regions are more likely to intend to have additional children than those in less vulnerable regions.

## 2.2 Environmental concerns and lower intention for childbearing

A confluence of many sorts of ecologically relevant consequences may be seen in the decision to live child-free with an environmental focus, which incorporates essential components of both individual environmentalism and environmental activism [37]. Szczuka [38] found that concerns about climate change were positively correlated with lower intended family sizes in Hungary and the Czech Republic. It indicates that environmental concerns influence family size preferences differently across countries, for instance, more broadly in Hungary and personally in the Czech Republic. In a different Chinese study, Li [39] showed a detrimental and substantial impact of air pollution on people's intention to have parenthood. Teenagers in the US who felt that the government should address environmental concerns were less likely to want to have children on average than those who disagreed [16]. Similarly, another research in the USA, Schneider-Mayerson and Leong [10], revealed that 59.8% of respondents reported being highly concerned about the carbon footprint of procreation. Additionally, younger respondents were more concerned about the climate impacts their children would experience than older respondents. Eissler et al. [40] showed that in sub-Saharan Africa, women in hotter climates want fewer children.

## 2.3 Environmental concerns and reproductive behaviour: Different perspectives

Research suggests that some individuals view childlessness as reducing the ecological footprint [41]. Sasser [42] explored the intersection between climate concerns and reproductive justice. She proposed an emerging perspective on racial health disparities and intergenerational justice. According to Fu et al. [43], the majority of young, educated Chinese people in China shared serious concerns about the long-term welfare of their (future) children in an environment that is changing. In Slovakia, Szczuka [38] revealed that a strong negative relationship was found, indicating that greater climate concerns were linked to preferences for smaller families in both general and on a personal level. Reproductive plans in Poland are impacted by worries about climate change and the fear of dying. Concerns about climate change cause positive reproductive intentions to decrease, but death anxiety causes them to grow [44]. Edmonston et al. [45] studied shifting fertility intentions in Canada by analyzing data from four national household surveys conducted in 1990, 1995, 2001, and 2006. The study revealed that Canadian women's reproductive intentions have been unchanged within a limited range over the past 16 years. Fearful of a catastrophic future environment, people may postpone or abstain from having children, lowering their ecological footprint [41]. Environmental concern and fertility plans are influenced by a variety of factors, including personality qualities, general views and ideas about gender roles, general life values concerning the importance of children or materialist and postmaterialist values, and demographic variables [46].

It is important to address the environmental, social, and psychological impacts of childlessness, particularly among young people who care about environmental sustainability. Although childlessness has been studied in the past [8,47], very few studies have explored the reasons and concerns of people considering childlessness in response to environmental issues. The present study aims to address the gap in the literature and open the door for further theoretical research and policy initiatives focused on understanding the variables that influence people's decisions to start or raise a family. This study argues that environmental problems in Bangladesh may have an impact on people's intention to have children in the future, even though most studies have examined whether environmental issues are related to childbearing intention. Thus, the study aims to investigate how environmental concerns among university students influence their future intentions to have children. It hypothesizes that there is a link between future childbearing intention and environmental concerns. It also hypothesizes that students who are concerned about the environment are more likely to desire children in the future, while planning to limit their intended family size.

## 3 Materials and methods

### 3.1 Research design and study area

The present study employs an explanatory research approach and quantitative techniques. The study location is Shah-jalal University of Science and Technology (SUST) in Sylhet, Bangladesh. SUST is Bangladesh's public research university [48]. Flooding occurred recently because of heavy rains [49]. Therefore, students were put in much more difficult conditions, and the experience of severe flooding has piqued the researchers' interest in investigating how they evaluate environmental concerns and childbearing intention.

### 3.2 Population and sampling

The study focused on SUST students enrolled in the master's program in 2020−2021 and the bachelor's program in 2017−2018. Applied Science and Technology, Life Science, Management and Business Administration, Physical Science, Agricultural and Mineral Science, and Social Science comprised SUST's academic programs. Students in the final year of the bachelor program (2017−18) and those in the master's program (2020−21) were selected as they may have more life experience than first-year students and may wish to marry after graduation. In addition, the master's and fourth-year students have completed the environmental studies program. They may have diverse views and be well-versed in environmental issues and fertility intentions. Four engineering departments- software, petroleum and mining, electrical and electronics, and mechanical- have not started master's programs. Therefore, there are no master's students in those departments.

Each departmental class representative knew the number of 4th-year undergraduate and master's students in the study. The departmental class representative reported 2386 students in academic sessions 2017−18 (4th-year undergraduate) and 2020−21 (master's program) in 28 departments. Of these 28 departments, 332 students were selected as a sample using Cochran's formula [50]. The representative sample size of the population from the normal distribution (Z = 1.96, 95% confidence level) was selected. P represents the required percentage. The researcher assumed p = 0.5, p + q = 1, q = p-1, and d = 0.05 intended error [50]. So, the calculated sample size is 332. Despite the calculated sample size of 332, the collected data came from 380 respondents. Simple random sampling was used to give each participant a fair chance of selection. This selection method ensures equal chances for all participants, resulting in an objective and unbiased sample [51]. Software known as the "Research Randomizer" [52] was used to select participants for the study.

### 3.3 Data collection techniques

This study employed a structured questionnaire and the social survey method [53] to gather data. To help with data gathering, the researchers recruited three field enumerators. Following an explanation of the objectives and key concepts of the research, the enumerators and study investigators gave the students a questionnaire to complete in class. Students had the chance to ask questions and received instructions on how to fill out the questionnaire. Each participant signed a permission form before data collection. Participants were guaranteed total anonymity and confidentiality of any information submitted in this permission form. Enrolling in this research carries no risk at all. It was ensured that participating in the research would have no negative consequences for them. The supplied information will only be used for research purposes. The authors aimed to share the questionnaire with at least 10 students from each department. However, the master's and fourth-year classes of the Forestry and Environmental Science (FES) under the discipline of Agriculture and Mineral Science, Physics and Chemical Engineering & Polymer Science departments under the discipline of Applied Science & Technology had ended. After that, the semester test of SUST started, so we could not collect data in the students' departments. Due to study pressure, students in the departments were not willing to respond. Students then usually go home and spend their winter holidays visiting family or travelling after exams. The researchers decided not to wait to collect data after the university opened because it would have been too late. However, as FES is the only department within

the discipline of Agriculture and Mineral Science and no students participated, the authors excluded this discipline from the final analysis.

The list of questions or statements was included in a questionnaire based on the relevant literature, particularly related to the environmental concern and fertility intentions [54]. The full-scale survey fieldwork followed a small pilot survey. To test whether respondents understood the adapted questions, a pilot survey was conducted from representative from each of the University's six faculties (Social Science, Physical Science, Agriculture & Mineral Science, Applied Science & Technology, Life Science, and Management & Business Administration) of the fourth-year undergraduate and master's degree students based on availability. This survey included 12 students. A pilot survey was conducted from October 20, 2022 to October 25, 2022. The researchers reevaluated the questions in light of the students' responses and points of view, adding any missing or recommended inquiries on future childbearing intention and environmental concerns. The revised questionnaire was discussed with the ethical approval committee. And the final data collection period was from November 19, 2022 to February 26, 2023. The completed questionnaire was then split into three sections (S1 Appendix): the sociodemographic data of the respondents, their future childbearing intention, and the impact of environmental concerns on future childbearing. Since Bengali is the primary language of the students, the researcher employed a back-translation process (first English, then Bengali) to guarantee the consistency and correctness of the questionnaire. This method also helped the students understand the questions. Before beginning data collecting, the project secured ethical permission from the SUST Research Ethics Board (SREB); the reference number for this approval is SREB/SS/SOC/PP 01 (2022).

## 3.4 Data analysis procedure

In this study, the dependent variable was future childbearing aspirations. Independent variables were chosen based on the literature (see Table 1) and included key sociodemographic factors such as gender, religion, disciplinary background, academic performance (cumulative grade point average [CGPA]), completion of environment-related academic courses, and whether students' home localities are vulnerable to climate change or EWEs. The researchers categorized respondents' responses to the question "impact of environmental concern on future childbearing" as a dichotomous scale. These items were measured on a dichotomous scale to capture a clear distinction between students who perceive environmental concerns as influencing their childbearing intentions and those who do not. This approach simplifies analysis and eases interpretation in statistical modeling. The survey responses were coded and analyzed using IBM SPSS Statistics (Version 26), and data visualization was conducted through R.

This study employed a stepwise analytical approach. First, descriptive statistics were used to understand the distribution of variables. Second, bivariate analysis (cross-tab with chi-square) explored the association between key variables. Finally, a multivariate regression model (binary logistic regression and Poisson regression) was conducted to assess the independent effects.

**Table 1. Item regarding impact of environmental concerns on future childbearing.**

| 1 | A harmful environment can endanger the health of the child |
|---|---|
| 2 | It is possible that childlessness has a beneficial effect on the environment. |
| 3 | Having more children in the future will increase environmental issues. |
| 4 | Going childless is a better way to help the environment than recycling. |
| 5 | Pollution will make a baby unhealthy. |
| 6 | Childfree lifestyle can reduce the effects of climate change. |
| 7 | People should consider having fewer children on the basis of environmental issues. |
| 8 | Having fewer children in the future is eco-friendly. |

## 3.5 Reliability and validity test

In this study, factor analysis was utilized to evaluate the convergent and discriminant validity, while Cronbach's Alpha was utilized to assess the reliability. Reliability is the degree to which an instrument is free from bias (error) and consistently assesses things throughout time and between various portions of the instrument [55]. A threshold over 0.6 in Cronbach's alpha analysis is frequently seen as appropriate for proving internal consistency dependability [56]. The study's instruments were deemed credible as all their Cronbach's alpha ratings were higher than the acceptable alpha threshold of 0.660 (Table 2).

Certain variables may impact the validity of the data. Convergent and discriminant validity are only a few of the various validity forums. To verify convergent validity, the researchers conducted a factor analysis and examined the factor loading pattern matrix. According to Carlson and Herdman [57], an instrument is acceptable if the average factor loading exceeds 0.7. Table 3 demonstrates that three components of the eight items on the impact of environmental concerns on future childbearing had an average factor loading higher than 0.7. This implies the validity of the study's instruments.

To evaluate discriminant validity, the square root of the average variance extracted (AVE) for each factor was compared to the correlations of other factors. Fornell & Larcker [58] state that discriminant validity is attained when the square root of the average variance extracted (AVE) is greater than the concept correlation. Three components of the eight items on the impact of environmental concerns on future childbearing had AVE higher than the correlation between the constructs, as Table 3 demonstrates. This implies the validity of the study's instruments.

## 4 Results

### 4.1 Descriptive statistics

The respondents' socio-demographic profile is in Fig 1. Of the students, 51.8% were male and 48.2% female. The majority of students were Muslim (81.1%), followed by Hindu (16.1%), Buddhist (1.3%), and Christian (.5%). Compared to the national statistics [23], religious distribution in the sample shows a slightly higher representation of minority groups. The gender distribution of respondents also reflects a balanced ratio, which is notable given that public university enrollment still slightly favours males. Students were asked about their Life Science, Social Science, Applied Science and Technology, Management and Business Administration, Physical Science, and Social Science backgrounds. The majority of students (39.5%) were in Social Science, followed by Applied Science and Technology (27.6%), Physical Science (19.7%),

**Table 2. Summary of reliability test.**

| Variable | No of items | No of sample | Cronbach's Alpha | Reliability |
|---|---|---|---|---|
| The impact of environmental concerns on future childbearing | 8 | 380 | 0.660 | Acceptable |

**Table 3. Summary of convergent and discriminant validity test.**

| Variable | Convergent Validity | No of items | Factor Analysis | | |
|---|---|---|---|---|---|
| The impact of environmental concerns on future childbearing | | | Component | Average Factor loading | |
| | | 8 | 3 | 0.792 | |
| | Discriminant Validity | 8 | | Average Variance Extracted | Component Correlation Matrix |
| | | | 1 | 0.677 | 0.060 |
| | | | 2 | 0.061 | 0.060 |
| | | | 3 | 0.652 | 0.060 |

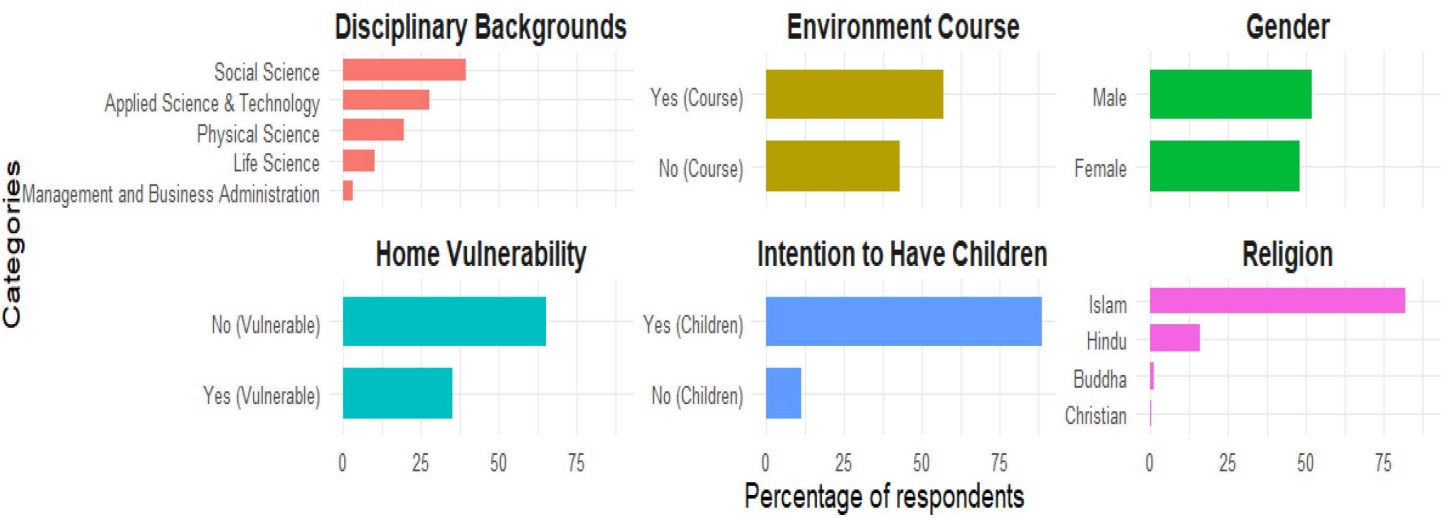

**Fig 1. Descriptive statistics.**

Life Science (10.0%), and Management and Business Administration (3.2%). At the University, 57.1% of students completed their environment-related academic course and 42.9% did not. Students were asked if their home location was vulnerable to climate change and extreme weather. The home areas of 65% of students are not at risk from climate change or EWEs, whereas 35% of students live in susceptible areas. They encountered a range of climate change phenomena and severe weather events in their local area, including flooding, extreme temperatures, earthquakes, heavy rainfall, pollution, drought, cyclones, river erosion, and storms. In comparison with US government estimates [25], the slightly lower percentage of students living in vulnerable areas are often less exposed to direct climate risks. Most students (88.7%) want children in the future, while 11.3% do not.

Fig 2 illustrates that most students want to have children in the future to experience parenthood, and 21.1% said they like children and that they are their greatest achievement. In other categories, respondents asserted that childbearing is an innate mechanism for ensuring the continuation of the human species, as well as for augmenting the available workforce and safeguarding future stability. Consequently, it is considered a moral obligation due to its divine nature. Students who do not want children in the future mention 5% dislike of children and 2.4% environmental concerns. In other categories, some common themes of respondents' statements were 'consideration of adoption', 'sense of responsibility', 'pessimism about the future', 'skepticism towards marriage', and 'concern about world suitability'.

Table 4 provides the continuous variable information. The table shows the mean CGPA of 380 respondents is 3.35, with a minimum CGPA of 2.75 to a maximum CGPA of 4.00. The mean number of children in the future of 337 respondents is 2.53, with a minimum value of 1 to a maximum of 5.

### 4.2 Respondents' understanding of socio-demographic factors, environmental concerns and future childbearing aspiration

To investigate the impact of environmental concerns on future childbearing aspirations, the authors employed the crosstab with Chi-square tests. Additionally, the Chi-square test was used to determine if there were any significant differences in socio-demographic factors that could potentially influence students' intentions to have children in the future.

**4.2.1 Association between items regarding the impact of environmental concerns on future childbearing and intention to have children in the future.** This research investigated the environmental variables that influence

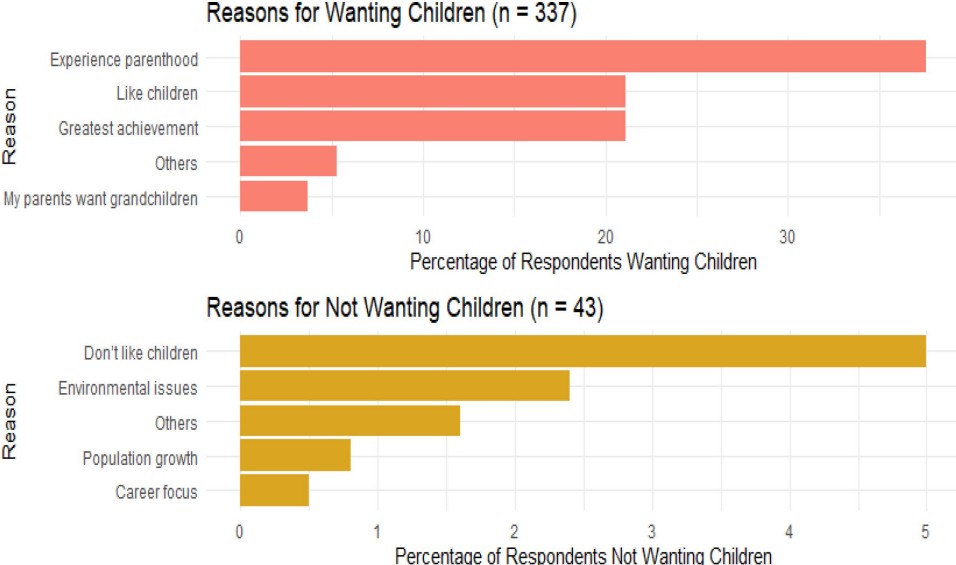

**Fig 2. Bar diagram showing the reasons behind wanting and not wanting children.**

childbearing aspirations. According to Fig 3, a chi-square test with a significance level of 0.05 was conducted to assess the impact of a detrimental environment on child health and future childbearing aspirations. The chi-square test yielded statistically significant results ($p < 0.05$). 89.1% of respondents who believed that an unfavorable environment could negatively impact the health of children expressed intentions to have children. Childlessness has a positive impact on the environment and is associated with a 1% increase in the intention to have children ($p = .000$). Out of the respondents, 94.2% who didn't believe that childlessness had a positive impact on the environment expressed intentions to have children. The findings also indicate that 93.4% of individuals who didn't believe that having additional children would exacerbate environmental problems were more inclined to have children. The chi-square test revealed a significant association between being childless having a positive impact on the environment, compared to recycling and the intention to have children ($p = .000$). There is a statistically significant relationship at a 1% level of significance ($p = .000$) between choosing a childfree lifestyle to mitigate climate change and the intention to have children in the future. Statistical analysis using the Chi-Square test revealed that there is no association between pollution negatively impacting the health of children and the intention to have children. Items about people should consider having fewer children based on environmental issues, and having fewer children is eco-friendly, are associated with intention to have children at 10% level of significance.

**4.2.2 Links between socio-demographic factors and intention to have children.** According to Table 5, a chi-square test with a significance level of 0.05 was conducted to assess gender and intention to have children. The chi-square test yielded statistically significant results ($p < 0.05$). 92.89% of male respondents who intend to have children in the future, while 15.85% of female respondents did not intend to have children in the future. In addition, 89.74% of Muslim

**Table 4. Continuous variable information.**

| Variable | N | Minimum | Maximum | Mean | Std. Deviation |
|---|---|---|---|---|---|
| CGPA of the respondents | 380 | 2.75 | 4.00 | 3.35 | .24 |
| Number of children in the future | 337 | 1 | 5 | 2.53 | 1.06 |

respondents wanted to have children in the future, while 86.89% Hindu, 60% Buddha, and 50% Christian respondents wanted to have children in the future at 10% level of significance. In terms of disciplinary backgrounds, 93.3% of social science students intend to have children for the future, whereas 91% of management and business administration, 88.5% of applied science and technology, 88% of physical science and 71% of life science background students aspire to have children at 5% level of significance. There is no significant association between factors such as completion of environment-related courses and perceived vulnerability of home locality to climate change or EWEs with intention to have children.

## 4.3 Significant factors among socio-demographic dimension, environmental concerns and future childbearing aspiration

In this study, the authors used Binary Logistic Regression (BLR) and Poisson Regression because findings from studies examining the link between environmental concerns and future childbearing aspirations are rare, both in Bangladesh and around the world. Another reason is that the dependent variable in BLR is a dichotomous one (intention to have children in the future: Yes = 1, No = 0), while the dependent variable in Poisson Regression is a count data (number of children in the future).

### 4.3.1 Significant factors affecting intention to have children in the future: BLR.
BLR was performed to assess the effect of various factors on the probability that respondents would report whether they expected to have children in the future. The model included several variables- covariates (gender, religion, disciplinary background, home locality

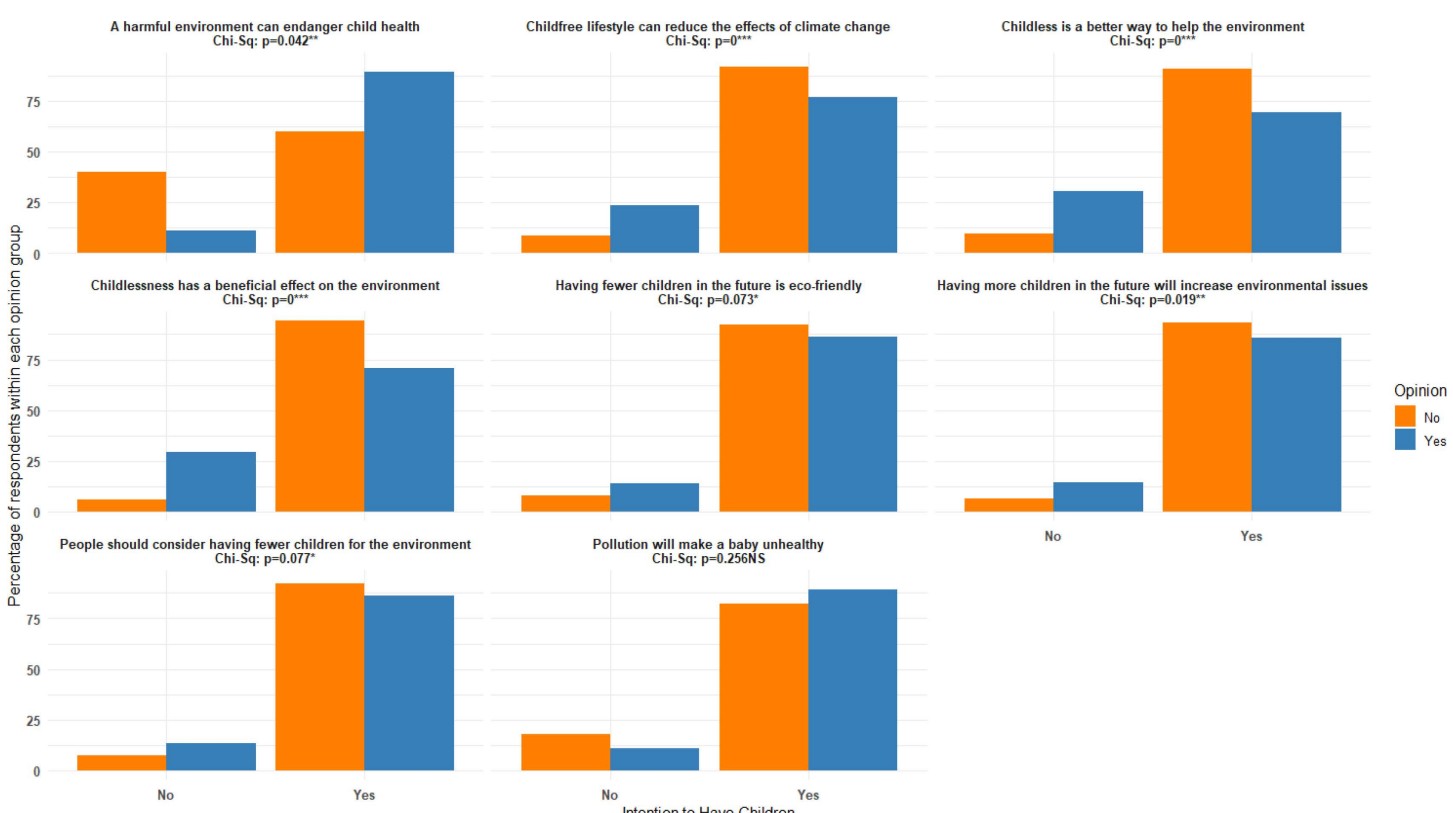

**Fig 3. Association between environmental concerns and future childbearing aspirations.**

vulnerable to EWEs or climate change, CGPA, environmental course completion), independent variable (item regarding the effect of environmental concerns on the intention to have future children) and dependent variable (intention to have children in the future). For selecting the impact of environmental concern on future childbearing as an independent variable, each item was analyzed separately to capture the specific aspects of concern due to the multidimensional nature of environmental concern. In addition, this item did not show a high Cronbach's Alpha value, therefore, independent analysis enables nuanced understanding of each concern instead of using it as a combined score.

The full model containing all predictors was highly significant, p < .005, indicating that it can differentiate between respondents who plan to have children in the future and those who do not. The model explained 17.5% (Cox and Snell $R^2$) and 34.5% (Nagelkerke $R^2$) of future childbearing and classified 90.3% with accuracy (Table 6).

Table 6 shows that two covariates (gender and disciplinary backgrounds) and one independent variable (childlessness has a beneficial effect on the environment) contribute to the model at 5% significance. Gender, disciplinary backgrounds (Applied Science and Technology, Management and Business Administration, Physical Science, Social Science), and childlessness have a beneficial effect on the environment were significant predictors. The strongest predictor of reporting intention to have children in the future was management & business administration backgrounds, recording an odds ratio of 23.93. It indicates that respondents who belong to management and business administration backgrounds were 23.93 times more likely to want children than life science students. Students in social science were 17.42 times more likely to want children than those in life science. Students in physical science were 9.23 times more likely to want children than those in life science. Students who believed childlessness had a positive impact on the environment were.139 times less likely to want children in the future than students who felt that childlessness did not have a positive impact on the environment. The odds ratio of the model also showed that women had a lower intention to have children than men.

**Table 5. Links between socio-demographic factors and intention to have children.**

| Variables | Intention to have children | | Total N (%) | Sig. |
|---|---|---|---|---|
| | No N (%) | Yes N (%) | | |
| **Gender of the respondents** | | | | .007** |
| Male | 14 (7.11%) | 183 (92.89%) | 197 (100%) | |
| Female | 29 (15.85%) | 154 (84.15%) | 183 (100%) | |
| **Religion of the respondents** | | | | .054* |
| Islam | 32 (10.26%) | 280 (89.74%) | 312 (100%) | |
| Hindu | 8 (13.11%) | 53 (86.89%) | 61 (100%) | |
| Buddha | 2 (40.00%) | 3 (60.00%) | 5 (100%) | |
| Christian | 1 (50.00%) | 1 (50.00%) | 2 (100%) | |
| **Disciplinary backgrounds** | | | | .004** |
| Life science | 11 (28.95%) | 27 (71.05%) | 38 (100%) | |
| Applied Science and Technology | 12 (11.43%) | 93 (88.57%) | 105 (100%) | |
| Management and Business Administration | 1 (8.33%) | 11 (91.67%) | 12 (100%) | |
| Physical science | 9 (12.00%) | 66 (88.00%) | 75 (100%) | |
| Social science | 10 (6.67%) | 140 (93.33%) | 150 (100%) | |
| **Completion of environment-related course** | | | | 0.26 |
| Yes | 28 (12.9%) | 189 (87.1%) | 217 (100%) | |
| No | 15 (9.2%) | 148 (90.8%) | 163 (100%) | |
| **Perceived vulnerability of home locality** | | | | 0.486 |
| Yes | 13 (9.77%) | 120 (90.23%) | 133 (100%) | |
| No | 30 (12.15%) | 217 (87.85%) | 247 (100%) | |

*p-value < 0.1; **p-value < 0.05.

The other covariates (i.e., religion; environmental-related course; CGPA; perceived vulnerability of home locality to EWEs and climate change) and independent variables (i.e., a harmful environment can endanger the health of the child; having more children in the future will increase environmental issues; going childless is a better way to help the environment than recycling; pollution will make a baby unhealthy; childfree lifestyle can reduce the effects of climate change; people should consider having fewer children based on environmental issues; having fewer children in the future is eco-friendly) were not statistically significant at the 5% level. But religion, CGPA, perceived vulnerability of home locality to climate change or EWEs, and item regarding going childless is a better way to help the environment than recycling was associated with intention to have children at a 10% level of significance. The odds ratio suggests that in terms of religion, respondents who belonged to Christianity were.019 times less likely to intend to have children in the future than respondents who belonged to Islam. In addition, students who felt that going childless is a better way to help the environment than recycling were.35 times less likely to intend to have children in the future than students who reported that going childless isn't a better way to help the environment than recycling. The odds ratio of CGPA suggests that for each one-unit increase in CGPA, the odds of intending to have children increased by a factor of 0.98. Respondents who perceive their home areas as vulnerable to climate change or EWEs were 2.21 times more likely to intend to have children in the future than those who do not perceive their home areas as vulnerable to climate change or EWEs.

**4.3.2 Significant factors affecting the number of children in the future: Poisson regression.** The researchers performed Poisson regression to examine the number of children in the future (dependent variable) based on some covariates (gender, home locality vulnerable to EWEs or climate change, CGPA, environmental course completion) and independent variables (items related to the impact of environmental concerns on future childbearing aspirations). However, Poisson regression was conducted only among respondents who reported intending to have children, as the follow-up question on the number of children in the future was asked conditionally (S1 Appendix). Therefore, individuals who stated they did not intend to have children were not included in the model. In addition, religion and disciplinary backgrounds were excluded from the Poisson regression model due to overdispersion. In the religion category, most of the respondents are Muslim, and in the disciplinary background category, most of the participants have a social science background. For this reason, these two variables were excluded from the Poisson regression analysis.

Table 7 displays the Poisson regression model for future child number and other independent variables. Pearson Chi square = .355 indicates under-dispersion and the model is suitable for this analysis. Omnibus Test p = .030 represents statistical significance. So, this model fulfilled the assumptions for conducting Poisson regression. The table also shows Poisson regression coefficient estimates and rate ratios. The table shows that only one independent variable (having more children will increase environmental issues) contributed to the model at 5% significance. The rate ratio for having more children will increase the environmental issues is .715. This indicated 28.5% decrease in the number of children in the future among those who thought that having more children in the future would increase environmental problems, compared to those who did not think that having more children in the future would increase environmental problems. The other explanatory variables are perceived vulnerability of home locality due to climate change or EWEs, a childfree lifestyle can reduce climate change effects, and having fewer children in the future is eco-friendly influencing number of children at a 10% level of significance.

# 5 Discussion

Very few prior studies sought to explain whether environmental issues directly impact individuals and their intention to have children in the future in the setting of Bangladesh. Prior research suggests that environmental variables are indirectly connected with reproductive aspirations. This study aims to address this gap by examining how environmental concerns and future childbearing aspirations are related to covariates like gender, religion, completion of environmental courses, academic performance, disciplinary backgrounds, and perceived vulnerability of home locality to EWEs or climate change, and how environmental concern itself influences these aspirations. Students from SUST, Bangladesh, were the study's

**Table 6. Parameter estimates for levels associated with intention to have children in the future.**

| Explanatory variables | Coefficient | Sig. | Odds ratio | 95% C.I. for EXP(B) | |
|---|---|---|---|---|---|
| | | | | Lower | Upper |
| **Gender** | | | | | |
| Female (1) <br> (Ref Male 0) | −1.290 | **.003\*\*** | .275 | .119 | .637 |
| **Religion** <br> (Ref Islam 1) | | **.081\*** | | | |
| Hindu (2) | −.354 | .469 | .702 | .269 | 1.830 |
| Buddha (3) | −.905 | .440 | .405 | .041 | 4.017 |
| Christian (4) | −3.960 | **.015\*\*** | .019 | .001 | .462 |
| **Disciplinary backgrounds** <br> (Ref Life science 1) | | **.001\*\*\*** | | | |
| Applied Science and Technology (2) <br> Management and Business | 1.523 | **.012\*\*** | 4.587 | 1.407 | 14.951 |
| Administration (3) | 3.175 | **.025\*\*** | 23.935 | 1.485 | 385.734 |
| Physical science (4) | 2.223 | **.002\*\*** | 9.232 | 2.308 | 36.928 |
| Social Science (5) | 2.858 | **.000\*\*\*** | 17.424 | 4.800 | 63.248 |
| **CGPA of the respondents** | 1.325 | **.098\*** | 3.762 | .783 | 18.081 |
| **Environmental-related course** | | | | | |
| Yes (1) <br> (Ref No 0) | −.310 | .488 | .733 | .305 | 1.760 |
| **Perceived vulnerability of home locality** | | | | | |
| Yes (1) <br> (Ref No 0) | .795 | **.074\*** | 2.215 | .926 | 5.298 |
| **A harmful environment can endanger the health of the child** | | | | | |
| Yes (1) <br> (Ref No 0) | 1.094 | .421 | 2.988 | .208 | 42.875 |
| **Childlessness has a beneficial effect on the environment** | | | | | |
| Yes (1) <br> (Ref No 0) | −1.972 | **.000\*\*\*** | .139 | .055 | .349 |
| **Having more children in the future will increase environmental issues** | | | | | |
| Yes (1) <br> (Ref No 0) | −.035 | .946 | .965 | .348 | 2.676 |
| **Going childless is a better way to help the environment than recycling** | | | | | |
| Yes (1) <br> (Ref No 0) | −1.050 | **.068\*** | .350 | .113 | 1.083 |
| **Pollution will make a baby unhealthy** | | | | | |
| Yes (1) <br> (Ref No 0) | .663 | .364 | 1.940 | .464 | 8.106 |
| **Childfree lifestyle can reduce the effects of climate change** | | | | | |
| Yes (1) <br> (Ref No 0) | −.158 | .763 | .854 | .307 | 2.375 |
| **People should consider having fewer children on the basis of environmental issues** | | | | | |
| Yes (1) <br> (Ref No 0) | −.130 | .805 | .878 | .311 | 2.476 |

*(Continued)*

**Table 6.** (Continued)

| Explanatory variables | Coefficient | Sig. | Odds ratio | 95% C.I. for EXP(B) | |
|---|---|---|---|---|---|
| | | | | Lower | Upper |
| **Having fewer children in the future is eco-friendly** | | | | | |
| Yes (1) (Ref No 0) | −.241 | .641 | .786 | .286 | 2.162 |

**Chi-square** = 72.98
**Sig**. = .000 (N = 380)
**R² (Cox & Snell)** = 17.5%; **R² (Nagelkerke)** = 34.5%
**Classification** = 90.3%

*p-value < 0.1; **p-value < 0.05; ***p-value < 0.01.

primary focus. The correlation between environmental issues and fertility intentions aligns with the local viewpoints of marginalized Bangladeshi communities in various regions [12,22,30].

First, based on the students' answers to how environmental concerns affect future childbearing, the authors examined the students' aspirations to have children. According to the cross-tabulation analysis using Chi-square tests, most of the items regarding the impact of environmental concerns on future childbearing significantly contribute to the explanation of the relationship between the intention to have children in the future. Items regarding the impact of environmental concerns on future childbearing (i.e., a harmful environment can endanger child health; childlessness has a beneficial effect on the environment, having more children in the future will increase environmental issues; childless is a better way to help the environment; childfree lifestyle can reduce the effects of climate change; people should consider having fewer children based on environmental issues; having fewer children in the future is eco-friendly) significantly explain the association of intention to have children in the future. The presence of pollution alone does not have a direct impact on students' intention to have children in the future. The analysis suggests that there is a link between environmental concerns and aspirations for future childbearing. A study conducted in Canada has yielded comparable findings regarding the notable relationship between environmental concerns and fertility intentions [1]. The study reveals that most of the students who took part in the study expressed apprehension regarding environmental issues and expressed their intention to have children in the future. In addition, the Chi-square test reveals that socio-demographic factors such as gender, religion, and disciplinary background have an impact on future childbearing aspirations. The results are consistent with the previous studies, such as Rackin et al. [16], which found similar trends among American adolescents, indicating that climate concerns are increasingly shaping reproductive intention in Western contexts. Similarly, Fu et al. [43] in young educated Chinese individuals also express a willingness to adjust their reproductive behavior in response to environmental degradation. The findings highlight that environmental concerns as a factor influencing fertility decisions might be across diverse cultural and socioeconomic contexts, indicating the need for further investigation.

Furthermore, the authors employed a BLR model to investigate the effects of the variables. The BLR analysis indicates that most of the predictors examined in the study had a concurrent impact on the intention to have children in the future. The analysis revealed that factors such as gender, religion, CGPA, disciplinary backgrounds and perceived vulnerability of home locality to climate change or EWEs were significantly associated with future childbearing intention. In addition, two specific items (i.e., "childlessness has a beneficial effect on the environment"; "going childless is a better way to help the environment than recycling") were also found to be statistically significant. The probability of students expressing a intention to have children in the future was higher if they met the following criteria: (a) identified as male, (b) belonged to the Islamic religion and had an academic background in social sciences, (c) had a high academic result (CGPA), and lived in a locality that is vulnerable to climate change or EWEs (d) did not believe that remaining childless has a positive impact on the environment (e) also did not consider that choosing not to have children is a more effective way to help the

**Table 7. Results of the multivariable generalized Poisson regression analysis to study the number of children in the future.**

| Explanatory Variables | Coefficient | Sig. | Rate Ratio | 95% Wald Confidence Interval for Exp(B) | |
|---|---|---|---|---|---|
| | | | | Lower | Upper |
| **Gender** | | | | | |
| Female (1) (Ref Male 0) | −.026 | .714 | .975 | .849 | 1.119 |
| **CGPA of the respondents** | −.134 | .371 | .875 | .652 | 1.173 |
| **Environmental-related course** | | | | | |
| Yes (1) (Ref No 0) | −.049 | .485 | .952 | .830 | 1.092 |
| **Perceived vulnerability of home locality** | | | | | |
| Yes (1) (Ref No 0) | .123 | **.092*** | 1.131 | .980 | 1.305 |
| **A harmful environment can endanger the health of the child** | | | | | |
| Yes (1) (Ref No 0) | −.199 | .550 | .819 | .427 | 1.574 |
| **Childlessness has a beneficial effect on the environment** | | | | | |
| Yes (1) (Ref No 0) | .047 | .631 | 1.048 | .865 | 1.271 |
| **Having more children in the future will increase environmental issues** | | | | | |
| Yes (1) (Ref No 0) | −.174 | **.034**** | .840 | .715 | .987 |
| **Going childless is a better way to help the environment than recycling** | | | | | |
| Yes (1) (Ref No 0) | 2.025E-5 | 1.000 | 1.000 | .755 | 1.325 |
| **Pollution will make a baby unhealthy** | | | | | |
| Yes (1) (Ref No 0) | −.039 | .772 | .961 | .736 | 1.255 |
| **Childfree lifestyle can reduce the effects of climate change** | .176 | **.083*** | 1.193 | .977 | 1.456 |
| Yes (1) (Ref No 0) | | | | | |
| **People should consider having fewer children on the basis of environmental issues** | | | | | |
| Yes (1) (Ref No 0) | | | | | |
| **Having fewer children in the future is eco-friendly.** | .050 | .606 | 1.052 | .868 | 1.274 |
| Yes (1) (Ref No 0) | −.181 | **.061*** | .834 | .690 | 1.009 |

**Dependent Variable:** Number of children in the future
**Omnibus Test:** Sig. = .030 (N = 337)
**Deviance:** Value/df = .355
**Pearson Chi-Square:** Value/df = .381

*p-value < 0.1; **p-value < 0.05; ***p-value < 0.01.

environment compared to recycling. This model also indicates that females exhibit a decreased tendency towards desiring a child in their future compared to males. The results are contradictory to the study conducted by Ewemooje et al. [31]. Regarding vulnerable areas, our findings align with Ahmed and Haq's [36] research, suggesting that climate vulnerability influences reproductive intention and should be considered in environmental policy. However, the result on disciplinary background contradicts those of Rovei et al. [33]. Additionally, the study result indicates that Muslim respondents are more likely to express an intention for future childbearing, also aligning with previously highlighting the role of religious beliefs in fertility preferences [34,35].

The Poisson regression analysis indicated that not all predictors had a simultaneous impact on the number of children in the future, although this effect was more pronounced among university students. Indeed, there is variation in the extent to which having fewer children in the future is considered eco-friendly. They strongly believed that having additional children in the future would worsen environmental problems in relation to population growth. The present study hypothesized that students who exhibit environmental concern are more likely to express an intention to limit the number of their future children. The Poisson regression model suggests that students are more likely to have limited their family size due to environmental concerns. The results align with Szczuka's [38] study, highlighting that climate concerns were significantly correlated with lower intended family sizes in Hungary and the Czech Republic. Additionally, the present research engages with Sasser's [42] critique of environmental narratives that place climate responsibility on individual reproductive choices. She argues that these choices should be understood within wider social and ecological systems, especially in marginalized contexts. Our findings contribute to this conversation by showing that students are more likely to plan for smaller families in the future due to concern for the environment. However, our data suggests that these beliefs are shaped by external forces such as environmental degradation and social pressures (i.e., religious beliefs, education and vulnerable conditions people face). In this sense, our research supports and extends Sasser's argument that reproductive issues are not only just driven by individual concerns but also shaped by wider contexts of environmental vulnerability. This research helps explain how environmental degradation in Bangladesh can shape people's future reproductive decisions and contributes to global debates on sustainable population policies and climate justice.

While this study focuses on the impact of environmental concern on future childbearing, it is important to acknowledge that other factors also play a crucial role. Personal finances, social support, partner preferences, career aspirations, cultural norms, and government policies can significantly influence the shaping of individual reproductive decisions, and future studies could explore how these factors interact with environmental concerns.

## 6 Limitations and recommendations for future research

The scope of our study is limited to Shahjalal University of Science and Technology. To obtain more precise generalizations, additional research is required that encompasses a wide range of university disciplines, considering students' intentions regarding future childbearing and the associated environmental concerns. Additionally, while the sample size was adequate for descriptive and bivariate analysis, it may be insufficient for complex multivariate models. This is an observational study and that no causal relation between specific concerns and childbearing intentions can be inferred. There could be non-causal mechanisms such as people with certain traits having correlated opinions and attitudes in the dimensions explored.

The risk of multicollinearity in such analysis may also affect the estimate stability. Although preliminary checks were conducted, future studies should use a larger sample size to address multicollinearity issues. Another limitation is that since our study relies on survey responses, participants might not answer accurately due to misunderstanding the question, and others might influence their views. This bias is particularly relevant in Bangladesh when discussing a sensitive or controversial topic. To ensure the clarity and consistency of the questionnaire, a pilot test was conducted with 12 participants. Based on their feedback, several questions were revised for clarity, relevance, and cultural sensitivity. This step was taken to minimize potential misunderstandings and ensure that participants interpreted the items consistently.

In addition, some complex variables, such as perceived vulnerability of home locality and the items regarding the impact of environmental concern on future childbearing, might not be accurately captured using a dichotomous variable. A more nuanced approach such as a Likert scale might better reflect the reality, which was not feasible in our study. The relationship between environmental concerns and future aspirations for childbearing is a unique concept. The variation in environmental concerns and future childbearing aspirations across different regions may lead to varying relationships influenced by distinct environmental issues and fertility intentions. There have been limited studies conducted recently on the relationship between environmental concerns and individuals' intention to have children in the future. Given the increasing severity of environmental issues, it is imperative to conduct further studies to better understand their impact on both the environment and future childbearing.

During the 1-year study, the data indicated that taking an environment-related course did not have a statistically significant impact on environmental concerns and the intention to have children in the future. Alternative outcomes may emerge following a complete implementation of the change, typically within a timeframe of 3–5 years. Hence, it is recommended to carry out further investigation over an extended duration. While developing countries face significant environmental risks, most studies focus on developed countries due to the availability of data on fertility and environmental issues in those regions. Future studies should prioritize examining the effects rather than focusing solely on the data. Additionally, it is important to carefully choose regions where the signs of environmental issues will become apparent in terms of fertility. The quantity of data gathered in our research amounts to a just 380. Further research could validate the conceptual framework and causal mechanisms established in this study by examining a more extensive sample size of university regions confronting diverse environmental challenges in developing nations that are susceptible to climate change. In addition, conducting a face-to-face interview will provide students with a deeper understanding of the new concept. This study model can be employed in developed nations to comprehend the links between environmental concern and its subsequent impact on future intention for having children. Outcomes and discoveries may vary between developed nations and developing nations. The results of this study could be useful in creating discussion topics for environmental scientists, which can aid reformers in comprehending possibilities for training, workshops, and the incorporation of data-validated models.

## 7 Conclusion

The environmental concerns are impacting the societal, economic, and cultural aspects of a country. The significance of comprehending the implications of environmental issues, such as fertility, on various population dynamics cannot be emphasized enough, especially in light of increasing global temperatures. Based on our research, environmental concerns are anticipated to have a diverse impact on students, as well as modify human reproductive practices and results. The study result indicates that students are concerned about environmental issues while also intending to have a child in the future. However, the findings also showed that participants with environmental concerns still want children, but they plan to limit their family size due to these concerns. The results may vary due to cultural, social, and political factors, highlighting the diversity of viewpoints on environmental issues, reproductive decision, and their intersection.

Our study reveals that environmental concerns have a substantial impact on individuals' aspirations for future childbearing. The findings of our study indicate that covariates such as gender, religion, CGPA, disciplinary background, and perceived vulnerability to home locality of EWEs or climate change are significantly associated with future childbearing aspirations. Women are considerably less likely to have children and plan to limit their family size in the future when environmental conditions worsen. Given Bangladesh's high population density, this study's findings provide insights into sustainable population management. The study result contributes to understanding the link between environmental concern and family planning, offering valuable implications for policies and balancing population growth with environmental sustainability. This study holds significant relevance for policy, research, and practices in Bangladesh, as it addresses the intersection of socio-cultural factors with population dynamics, environmental sustainability, and equitable development.

## Supporting information

**S1 Appendix. Questionnaire.** The questionnaire was used for data collection.
(DOC)

## Acknowledgments

We want to convey our heartfelt thanks to the respondents for participating in this study. A special thanks to the data enumerators for their tremendous support during fieldwork and transcription. Without the assistance, we would not have been able to conduct our fieldwork so precisely.

## Author contributions

**Conceptualization:** Shah Md Atiqul Haq.

**Data curation:** Bijoya Saha.

**Formal analysis:** Bijoya Saha, Shah Md Atiqul Haq.

**Investigation:** Bijoya Saha.

**Methodology:** Bijoya Saha, Shah Md Atiqul Haq.

**Software:** Bijoya Saha.

**Supervision:** Shah Md Atiqul Haq.

**Validation:** Bijoya Saha.

**Writing – original draft:** Bijoya Saha, Shah Md Atiqul Haq.

**Writing – review & editing:** Bijoya Saha, Shah Md Atiqul Haq.

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
