## [Decision Letter · Decision Letter 0]

20 Jan 2025

PONE-D-24-46311The nexus between environmental concern and future childbearing aspirations among university students in BangladeshPLOS ONE

Dear Dr. Atiqul Haq,

Thank you for submitting your manuscript to PLOS ONE. After careful consideration, we feel that it has merit but does not fully meet PLOS ONE’s publication criteria as it currently stands. Therefore, we invite you to submit a revised version of the manuscript that addresses the points raised during the review process.

Three experts have reviewed the manuscript noting a number of limitations on the statistical reporting and the existence of unwarranted claims in the introduction and discussion that should be removed or modified. In addition, the article would benefit from going through the Strobe checklist (https://www.strobe-statement.org/download/strobe-checklist-cross-sectional-studies-pdf) to ensure that all technical aspects have been addressed. The structure also should be enhanced: there are separate analysis on which the rationale and the rationale of the sequence is not clear. Also regarding whether to use a scale vs all dychotomous items. The methods section and the research objectives should be made clearer so that the analysis flows from these. There is also a problem on sample size that might not be adequate for some of the more sophisticated analysis. As you know, due to multicollinearity, variances in multiple variable methods are higher that for simple two-variable analysis that you have used to ascertain sample size. This should be acknowledged as a limitation.

We look forward to receiving your revised manuscript.

Kind regards,

José Antonio Ortega, Ph.D.

Academic Editor

PLOS ONE

Journal Requirements:

2. We note that your Data Availability Statement is currently as follows: [Because respondents were notified that their information would be published, the datasets collected and/or analyzed during the current study are available as supplementary file.]

Please confirm at this time whether or not your submission contains all raw data required to replicate the results of your study. Authors must share the “minimal data set” for their submission. PLOS defines the minimal data set to consist of the data required to replicate all study findings reported in the article, as well as related metadata and methods (https://journals.plos.org/plosone/s/data-availability#loc-minimal-data-set-definition ).

If your submission does not contain these data, please either upload them as Supporting Information files or deposit them to a stable, public repository and provide us with the relevant URLs, DOIs, or accession numbers. For a list of recommended repositories, please see https://journals.plos.org/plosone/s/recommended-repositories .

3. We note that Figure 1 in your submission contain [map/satellite] images which may be copyrighted. All PLOS content is published under the Creative Commons Attribution License (CC BY 4.0), which means that the manuscript, images, and Supporting Information files will be freely available online, and any third party is permitted to access, download, copy, distribute, and use these materials in any way, even commercially, with proper attribution. For these reasons, we cannot publish previously copyrighted maps or satellite images created using proprietary data, such as Google software (Google Maps, Street View, and Earth). For more information, see our copyright guidelines: http://journals.plos.org/plosone/s/licenses-and-copyright .

We recommend that you contact the original copyright holder with the Content Permission Form (http://journals.plos.org/plosone/s/file?id=7c09/content-permission-form.pdf ) and the following text:

“I request permission for the open-access journal PLOS ONE to publish XXX under the Creative Commons Attribution License (CCAL) CC BY 4.0 (http://creativecommons.org/licenses/by/4.0/ ). Please be aware that this license allows unrestricted use and distribution, even commercially, by third parties. Please reply and provide explicit written permission to publish XXX under a CC BY license and complete the attached form.”

Reviewers' comments:

Reviewer's Responses to Questions

**Comments to the Author**

1. Is the manuscript technically sound, and do the data support the conclusions?

Reviewer #1: Partly

Reviewer #2: Partly

Reviewer #3: Yes

2. Has the statistical analysis been performed appropriately and rigorously? 

Reviewer #1: I Don't Know

Reviewer #2: Yes

Reviewer #3: I Don't Know

3. Have the authors made all data underlying the findings in their manuscript fully available?

Reviewer #1: Yes

Reviewer #2: Yes

Reviewer #3: Yes

4. Is the manuscript presented in an intelligible fashion and written in standard English?

Reviewer #1: Yes

Reviewer #2: Yes

Reviewer #3: Yes

5. Review Comments to the Author

Reviewer #1: This paper presents interesting primary research that explores the nexus between environmental concerns and future childbearing aspirations among university students in Bangladesh. The author seeks to address an appropriate gap in the literature on this topic given this question has been underexplored in low- and middle-income settings, including Bangladesh.

1. Referencing and assumptions: several claims are made in the text without an adequate source for the information. For example:

Line 47: Despite there being a reference for this, the author should clarify the location of Davis et al.’s study to avoid confusion that this statement is generalisable to all women.

Line 56: The statement “according to research, having children will damage the ecosystem” needs to be clearer on what research the author is referring to and this should be listed as a citation.

Line 96: The statement that university students in Bangladesh “are more aware of family planning and concerned about the environment” needs to be backed up by a citation.

Line 143/44: The authors appear to advocate for a viewpoint which risks introducing personal bias. The authors should clarify why this perspective is being promoted and ensure framing of items do not reflect bias.

Line 584: The statement “women in the country may exhibit lower levels of ambition” is not supported by a source. It is unclear what ‘ambition’ means in this context, so further clarification is necessary.

I recommend that the authors review their manuscript to ensure that all assertions are appropriately supported with references and any subjective viewpoints should be clearly distinguished from evidence-based conclusions.

2. Structure and readability: while the use of sub-headings contributes to the logical flow of the paper, the structure within sections – particularly the introduction and literature review – could be improved. For example, lines 121-139 could be organised by dividing studies into those that found environmental concerns led to lower desires to have children vs. those that found the opposite conclusion. Language improvements are also needed. Typos, grammar issues, and repetition currently affect readability of the study. I have listed a few examples of each below but not an extensive list for the purposes of brevity. I would encourage the authors to thoroughly review the paper, and collaboration with a writing coach or copyeditor may be beneficial if possible.

Line 39: Arnocky spelt incorrectly.

Line 58-60: Sentence structures are unclear.

Line 207: Typo for ‘participating’

Lines 234/235: Repeated sentence.

3. Definitions and terminology: Key terms should be clearly defined and used consistently. This is particularly important when referring to childbearing aspirations/objectives/desires/intentions/goals which are all used throughout the manuscript, are undefined, and may imply very different meanings. For example, the author could consider the Traits-Desires-Intentions-Behaviours Model (Miller 1994; 2011) which explores the difference between ‘desire’ and ‘intention’ in the context of reproductive decision-making. The authors ought to either use consistent terminology in this case and refer to the same term that was used on the survey to participants, or be really clear on different definitions if using multiple terms. Definitions should also be delineated for how the authors are interpreting ‘environmental concern’. Finally, acronyms need to be expanded when first used in the text which applies to line 210 and line 246.

4. Methodology/Results: certain elements of the methodology and results sections require greater clarification. Examples are listed below:

There ought to be a reference for the ‘Research Randomizer’ tool.

Line 270-272 and 278-281: the authors state that only three out of eight questions met the necessary criteria. How does this validate the tools if less than 50% were satisfied?

Line 290: gender percentage is incorrectly reported in the text (correct to 48.2% aligned with Table 5).

Line 296: this is the only time Agricultural and Mineral Science is referred to as a participant’s degree. This puts the percentage for degrees off by +0.5%. The authors should clarify if this was a participant/s degree or not and adjust the text and table 5 accordingly.

The survey questions ought to be included in the appendix to give readers insight into the tools used.

5. Limitations: the limitations section could be expanded to include, for example, self-reporting bias inherent to surveys, and the issues with using binary measures for complex variables e.g., classifying homes as vulnerable or not vulnerable to climate change (line 242).

6. Discussion: The discussion would benefit from further exploration of the broader range of factors influencing childbearing decisions. Whilst the study is focussed on the impact of environmental concern, other considerations – such as personal finances, partner preferences, social support etc. – are also likely to influence student’s opinions so it is worth addressing this, even if briefly in a couple of sentences.

7. Conclusion: While the data supports the conclusions made, the framing that students who are environmentally concerns are more likely to want children, yet want fewer of them is confusing. It should be clarified that while participants with environmental concerns may still want children, the findings also showed that they plan on limiting family size due to these concerns. This framing should also be addressed in the abstract (line 24). Additionally, the conclusion should be limited to a concise summary of the study which avoids the introduction of new ideas.

Reviewer #2: Introduction

Avoid overusing "laundry lists" when constructing paragraphs, as they can obscure the main message the author wants to convey. For example, in lines 75–79, it would be more effective to summarize the points about family planning in 1–2 concise sentences.

Line 81:The introduction of the Bangladesh context feels abrupt and disconnected in this paragraph. Consider providing context first—such as explaining why densely populated countries might be more impacted by future childbearing aspirations. Then, introduce Bangladesh as an example, along with other similarly populated countries, to offer a more comprehensive perspective.

Line 96: “The question, “How do university students in Bangladesh, who are more aware of family planning and concerned about the environment, make their decisions about having children in the future?” lacks sufficient explanation and justification. It is unclear why university students are presumed to be more aware of family planning or have heightened environmental concerns. Providing data or references to support these assumptions would strengthen the argument and improve clarity.

Methods and Results

Line 234-235: repeated sentences.

Can you clarify whether you are using factor analysis or component analysis? Based on the wording, "Component analysis extraction technique" likely refers to principal component analysis (PCA). However, PCA is a data reduction technique and not ideal for establishing convergent validity compared to factor analysis.

Line 269: The phrase "average factor loading correlation" is unconventional. Factor loadings and correlations are distinct concepts.

Is there any specific reason to analyze the environmental concerns variable items independently? Please elaborate.

Discussion,

Line 492: How do the data of Americans and Chinese relate to the findings?

A more thorough review in the introduction will enable the authors to produce a more empirically and theoretically informed discussion.

Overall

Many variables are examined in the study; however, the lack of a comprehensive literature review limits the depth of the discussion on the findings. For instance, in lines 506–508, how can the increased willingness to have children among respondents with specific demographic characteristics be explained? For example, Muslim respondents are more likely to express a desire for future childbearing. Could this tendency be related to their religious beliefs?

Reviewer #3: Thank you for letting me read what seems to be a methodologically stringent and sound study. Before being publishable, I nonetheless think the manuscript needs to strengthen its relevance for both theory and practice. I have some suggestions for this at the end of my comments.

The very first paragraph is about environmental concerns in general, resulting in a bit vague introduction. If the topic is how such concerns are related to fertility intentions, then this (including why this topic is important) should be presented already in the first paragraph.

The description of the Bangladesh context is useful, and should perhaps have its own header? For instance, “Context of the study”.

I am not very experienced with quantitative methods, but I nonetheless did not catch a rationale for having dichotomous items for the variable “The impact of environmental concerns on future childbearing”. Beyond this, my comments are indeed restricted by the fact that I am not well trained to see limitations in the described procedures and statistical analyses.

What I do think this manuscript is lacking, is a clearer idea of how and why the results are relevant. For instance, I do not think the statement “This study plays an important role in attaining the Sustainable Development Goals (SDGs) in Bangladesh” (p. 34-35) is justified by the manuscript. The relation to the SDGs has to be elaborated on, and I also think the manuscript should go in dialogue with the article “At the intersection of climate justice and reproductive justice” (Sasser, 2024, https://doi.org/10.1002/wcc.860).

6. PLOS authors have the option to publish the peer review history of their article (what does this mean? ). If published, this will include your full peer review and any attached files.

**Do you want your identity to be public for this peer review?** For information about this choice, including consent withdrawal, please see our Privacy Policy .

Reviewer #1: No

Reviewer #2: No

Reviewer #3: No

---

## [Author Response · Author response to Decision Letter 1]

11 Mar 2025

Comments to the Author

Reviewer #1: This paper presents interesting primary research that explores the nexus between environmental concerns and future childbearing aspirations among university students in Bangladesh. The author seeks to address an appropriate gap in the literature on this topic given this question has been underexplored in low- and middle-income settings, including Bangladesh.

1. Referencing and assumptions: several claims are made in the text without an adequate source for the information. For example:

Line 47: Despite there being a reference for this, the author should clarify the location of Davis et al.’s study to avoid confusion that this statement is generalisable to all women.

Response: Thank you for the comment. The confusion has been cleared.

Line 56: The statement “according to research, having children will damage the ecosystem” needs to be clearer on what research the author is referring to and this should be listed as a citation.

Response: The statement has been corrected and added reference.

Line 96: The statement that university students in Bangladesh “are more aware of family planning and concerned about the environment” needs to be backed up by a citation.

Response: Citations have been added and also added justification.

Line 143/44: The authors appear to advocate for a viewpoint which risks introducing personal bias. The authors should clarify why this perspective is being promoted and ensure framing of items do not reflect bias.

Response: The confusion has been cleared.

Line 584: The statement “women in the country may exhibit lower levels of ambition” is not supported by a source. It is unclear what ‘ambition’ means in this context, so further clarification is necessary.

Response: The statement has been removed as it is not supported in the manuscript.

I recommend that the authors review their manuscript to ensure that all assertions are appropriately supported with references and any subjective viewpoints should be clearly distinguished from evidence-based conclusions.

2. Structure and readability: while the use of sub-headings contributes to the logical flow of the paper, the structure within sections – particularly the introduction and literature review – could be improved. For example, lines 121-139 could be organised by dividing studies into those that found environmental concerns led to lower desires to have children vs. those that found the opposite conclusion. Language improvements are also needed. Typos, grammar issues, and repetition currently affect readability of the study. I have listed a few examples of each below but not an extensive list for the purposes of brevity. I would encourage the authors to thoroughly review the paper, and collaboration with a writing coach or copyeditor may be beneficial if possible.

Response: Thank you for your suggestion. The sub-sections have been added in the literature section.

Line 39: Arnocky spelt incorrectly.

Response: Corrected

Line 58-60: Sentence structures are unclear.

Response: Corrected

Line 207: Typo for ‘participating’

Response: Corrected

Lines 234/235: Repeated sentence.

Response: Corrected

3. Definitions and terminology: Key terms should be clearly defined and used consistently. This is particularly important when referring to childbearing aspirations/objectives/desires/intentions/goals which are all used throughout the manuscript, are undefined, and may imply very different meanings. For example, the author could consider the Traits-Desires-Intentions-Behaviours Model (Miller 1994; 2011) which explores the difference between ‘desire’ and ‘intention’ in the context of reproductive decision-making. The authors ought to either use consistent terminology in this case and refer to the same term that was used on the survey to participants, or be really clear on different definitions if using multiple terms. Definitions should also be delineated for how the authors are interpreting ‘environmental concern’. Finally, acronyms need to be expanded when first used in the text which applies to line 210 and line 246.

Response: The term “childbearing aspirations” means aspiration or intention to have children. So, the use of terms such as: childbearing aspirations and environmental concern are interpreted according to the study. The authors use consistent terminology such as aspiration/ intention to have children. The definitions of the term have also been included. And acronyms have been expanded: Physics, Forestry and Environmental Science, Chemical Engineering & Polymer Science, Extreme Weather Events.

4. Methodology/Results: certain elements of the methodology and results sections require greater clarification. Examples are listed below:

There ought to be a reference for the ‘Research Randomizer’ tool.

Response: Reference has been added

Line 270-272 and 278-281: the authors state that only three out of eight questions met the necessary criteria. How does this validate the tools if less than 50% were satisfied?

Response: The confusion has been cleared.

Line 290: gender percentage is incorrectly reported in the text (correct to 48.2% aligned with Table 5).

Response: Corrected. You can see the changes in the results section.

Line 296: this is the only time Agricultural and Mineral Science is referred to as a participant’s degree. This puts the percentage for degrees off by +0.5%. The authors should clarify if this was a participant/s degree or not and adjust the text and table 5 accordingly.

The survey questions ought to be included in the appendix to give readers insight into the tools used.

Response: The confusion has been cleared. The Department of Forestry and Mineral Science, under the Faculty of Agricultural and Mineral Science, was included in the survey. However, as it is the only department within this faculty and only one student participated—due to the master's and fourth-year classes having ended and the semester exams having started during data collection. Therefore, we excluded this department from our analysis. Also, the survey question has been included.

5. Limitations: the limitations section could be expanded to include, for example, self-reporting bias inherent to surveys, and the issues with using binary measures for complex variables e.g., classifying homes as vulnerable or not vulnerable to climate change (line 242).

Response: Limitation section has been expanded according to the your suggestion.

6. Discussion: The discussion would benefit from further exploration of the broader range of factors influencing childbearing decisions. Whilst the study is focussed on the impact of environmental concern, other considerations – such as personal finances, partner preferences, social support etc. – are also likely to influence student’s opinions so it is worth addressing this, even if briefly in a couple of sentences.

Response: Added in discussion section

7. Conclusion: While the data supports the conclusions made, the framing that students who are environmentally concerns are more likely to want children, yet want fewer of them is confusing. It should be clarified that while participants with environmental concerns may still want children, the findings also showed that they plan on limiting family size due to these concerns. This framing should also be addressed in the abstract (line 24). Additionally, the conclusion should be limited to a concise summary of the study which avoids the introduction of new ideas.

Response: Conclusion and abstract section have been modified.

Reviewer #2: Introduction

Avoid overusing "laundry lists" when constructing paragraphs, as they can obscure the main message the author wants to convey. For example, in lines 75–79, it would be more effective to summarize the points about family planning in 1–2 concise sentences.

Response: Thank you for comment. The sentences have been removed as it is not well-fitted in the manuscript.

Line 81: The introduction of the Bangladesh context feels abrupt and disconnected in this paragraph. Consider providing context first—such as explaining why densely populated countries might be more impacted by future childbearing aspirations. Then, introduce Bangladesh as an example, along with other similarly populated countries, to offer a more comprehensive perspective.

Response: Thank you for your suggestion. You can see the changes: Introduction section has been split and a new heading has been added, “context of the study: Bangladesh

Line 96: “The question, “How do university students in Bangladesh, who are more aware of family planning and concerned about the environment, make their decisions about having children in the future?” lacks sufficient explanation and justification. It is unclear why university students are presumed to be more aware of family planning or have heightened environmental concerns. Providing data or references to support these assumptions would strengthen the argument and improve clarity.

Response: Thank you for your comment. The statement has been modified and provided reference as well.

Methods and Results

Line 234-235: repeated sentences.

Response: Corrected

Can you clarify whether you are using factor analysis or component analysis? Based on the wording, "Component analysis extraction technique" likely refers to principal component analysis (PCA). However, PCA is a data reduction technique and not ideal for establishing convergent validity compared to factor analysis.

Response: Corrected. You can see the changes in the reliability and validity test.

Line 269: The phrase "average factor loading correlation" is unconventional. Factor loadings and correlations are distinct concepts.

Response: The confusion has been cleared.

Is there any specific reason to analyze the environmental concerns variable items independently? Please elaborate.

Response: For selecting the impact of environmental concern on future childbearing as an independent variable: each item was analyzed separately to capture the specific aspects of concern due to the multidimensional nature of environmental concern. In addition, this item did not show a high Cronbach's Alpha value, therefore, independent analysis enables nuanced understanding of each concern instead of using it as a combined score. This justification has also been included in the manuscript.

Discussion,

Line 492: How do the data of Americans and Chinese relate to the findings?

A more thorough review in the introduction will enable the authors to produce a more empirically and theoretically informed discussion.

Response: Corrected. You can see the changes in the discussion section.

Overall

Many variables are examined in the study; however, the lack of a comprehensive literature review limits the depth of the discussion on the findings. For instance, in lines 506–508, how can the increased willingness to have children among respondents with specific demographic characteristics be explained? For example, Muslim respondents are more likely to express a desire for future childbearing. Could this tendency be related to their religious beliefs?

Response: Thank you for your suggestion. Specific demographic variables were examined in the study, and these variables have been included in the literature review section. You can see the changes in “Factors associated with fertility decision making” and “Discussion” section.

Reviewer #3: Thank you for letting me read what seems to be a methodologically stringent and sound study. Before being publishable, I nonetheless think the manuscript needs to strengthen its relevance for both theory and practice. I have some suggestions for this at the end of my comments.

1. The very first paragraph is about environmental concerns in general, resulting in a bit vague introduction. If the topic is how such concerns are related to fertility intentions, then this (including why this topic is important) should be presented already in the first paragraph.

The description of the Bangladesh context is useful, and should perhaps have its own header? For instance, “Context of the study”.

Response: Thank you for your suggestion. You can see the changes in Introduction. A header “Context of the study Bangladesh” has been added.

2. I am not very experienced with quantitative methods, but I nonetheless did not catch a rationale for having dichotomous items for the variable “The impact of environmental concerns on future childbearing”. Beyond this, my comments are indeed restricted by the fact that I am not well trained to see limitations in the described procedures and statistical analyses.

Response: Thank you for raising this issue. We acknowledge that a Likert scale could have provided more nuanced insights into the degree of influence. We have also addressed this concern in the limitation section. However, we choose a dichotomous scale to capture a clear distinction between individuals who perceive environmental concerns as influencing their childbearing decisions and those who do not. This approach simplifies analysis and eases of interpretation in statistical modeling.

3. What I do think this manuscript is lacking, is a clearer idea of how and why the results are relevant. For instance, I do not think the statement “This study plays an important role in attaining the Sustainable Development Goals (SDGs) in Bangladesh” (p. 34-35) is justified by the manuscript. The relation to the SDGs has to be elaborated on, and I also think the manuscript should go in dialogue with the article “At the intersection of climate justice and reproductive justice” (Sasser, 2024, https://doi.org/10.1002/wcc.860).

Response: Thank you for suggestion. We remove the statement “This study plays an important role in attaining the Sustainable Development Goals (SDGs) in Bangladesh” because it is not well-fitted for the manuscript. However, we cited the Sasser (2024) work in the literature section and incorporate relevant discussions on climate justice and reproductive justice related to our findings, you can see in the discussion section.

---

## [Decision Letter · Decision Letter 1]

28 Mar 2025

PONE-D-24-46311R1The nexus between environmental concern and future childbearing aspirations among university students in BangladeshPLOS ONE

Dear Dr. Atiqul Haq,

Thank you for submitting your manuscript to PLOS ONE. After careful consideration, we feel that it has merit but does not fully meet PLOS ONE’s publication criteria as it currently stands. Therefore, we invite you to submit a revised version of the manuscript that addresses the points raised during the review process.

The comments of the editor have not been addressed. I reiterate:

“The structure also should be enhanced: there are separate analysis on which the rationale and the rationale of the sequence is not clear. Also regarding whether to use a scale vs all dychotomous items. The methods section and the research objectives should be made clearer so that the analysis flows from these. There is also a problem on sample size that might not be adequate for some of the more sophisticated analysis. As you know, due to multicollinearity, variances in multiple variable methods are higher that for simple two-variable analysis that you have used to ascertain sample size. This should be acknowledged as a limitation.”

In addition, citation style is not according to PLOS ONE numbered style. This needs to be fixed.

The questionnaire, currently included as an appendix to the main manuscript, should be provided as a separate file.

Regarding the first reviewer, who was unavailable, apparently the points have been addressed. The second reviewer provides some optional suggestions. Reviewer 3 indicates that language must be improved. The editor concurs on the need to copy-edit the manuscript.

Examples of issues that should be addressed, but there are many. The whole writing up needs to be improved.

L. 83: The reading is now very different. “Highly productive” does not seem to apply given the location of Bangladesh in GDP rankings. The beautifulness of the rivers is also a subjective term such as those reviewer 1 was indicating in the previous revision.L. 196-198: Remove consideration on the buildings of SUST unless it is deemed relevant for some reason. Discussion should be based on academic program and level.L. 245: literature should be singular.Example of needed rephrasing is “measurement and analysis”. Some issues with language: “The researchers coded the following predictors into269 categorical variables for binary logistic regression and other analyses:”. Why mixing the analysis with the coding? Was the coding ex-post carried out by the researchers as this sentence implies? It seems this was a closed questionnaire so that you should not mention “coding” at all. More important: we do not need to know the numbers of the codes at all! Also, on this: “which faculty 272 members are currently studying” you mean “field of study”. As commented, grammar needs to be revised and improved.L. 278: What is meant by “graphical distributions were created using RStudio”. You mean graphics were produced with R (RStudio is the gui. “Graphical distributions” is meaningless. Statistical distributions are not graphical.)Figure 3: The axis should be clearly identified: percentage what?Table 5 should include the respective Ns as a new column.Table 6 is missing NL. 467 and later: It is not specified whether the poisson regression includes also those reporting not wanting to have children codified as 0 children. It should, otherwise you are not dealing with the complete distribution.For discussion: You are including as a variable “Vulnerability to home locality” (which probably should be “vulnerability of”. Note this is student’s perception. It might well be that students living in the same locality fill this up differently.

In general, as commented, there are still many problems of structure and language. If these have not been amended the manuscript cannot be considered for publication.

We look forward to receiving your revised manuscript.

Kind regards,

José Antonio Ortega, Ph.D.

Academic Editor

PLOS ONE

Reviewers' comments:

Reviewer's Responses to Questions

**Comments to the Author**

1. If the authors have adequately addressed your comments raised in a previous round of review and you feel that this manuscript is now acceptable for publication, you may indicate that here to bypass the “Comments to the Author” section, enter your conflict of interest statement in the “Confidential to Editor” section, and submit your "Accept" recommendation.

Reviewer #2: All comments have been addressed

Reviewer #3: (No Response)

2. Is the manuscript technically sound, and do the data support the conclusions?

Reviewer #2: Yes

Reviewer #3: Yes

3. Has the statistical analysis been performed appropriately and rigorously? 

Reviewer #2: Yes

Reviewer #3: I Don't Know

4. Have the authors made all data underlying the findings in their manuscript fully available?

Reviewer #2: Yes

Reviewer #3: Yes

5. Is the manuscript presented in an intelligible fashion and written in standard English?

Reviewer #2: Yes

Reviewer #3: No

6. Review Comments to the Author

Reviewer #2: Context of Study: While the paper provides an environmental description of Bangladesh, it would be beneficial to include a more detailed demographic context, particularly characteristics relevant to the studied participants (e.g., religious beliefs, education levels, socioeconomic status). Additionally, comparing these demographics to national data would help illustrate the representativeness of the sample and strengthen the study’s generalizability.

Limitations: Has any effort been made to ensure that respondents interpreted the questions consistently? While it is commendable that the study acknowledges its limitations, it would be helpful to outline any steps taken to minimize potential misunderstandings, such as piloting the survey, providing clarifications, or conducting follow-up checks. This would enhance the study’s methodological rigor.

Reviewer #3: I appreciate the work you have put in to respond to all three reviewers, and I think your manuscript is a lot better now. My two main concerns now are language and the use of Sasser (2024).

- Language: I find parts of the manuscript to be a bit hard to read, as many sentences are long and sometimes confusing. You should go through the whole manuscript trying to make the language clearer. Example: "The analysis revealed that factors such as gender, religion, CGPA, disciplinary backgrounds, home locality susceptible to climate change or EWEs, childlessness have a beneficial effect on the environment, and going childless is a better way to help the environment than recycling, found to be statistically significant." (p. 30, lines 536-540). Furthermore, the last two factors listed here seem more like items to me?

- Sasser (2024): You write that you have tried to "incorporate relevant discussions on climate justice and reproductive justice related to our findings". The way I read your discussion, you refer briefly to Sasser (2024) without actually discussing how your findings relate to her arguments. This must be fixed.

7. PLOS authors have the option to publish the peer review history of their article (what does this mean? ). If published, this will include your full peer review and any attached files.

**Do you want your identity to be public for this peer review?** For information about this choice, including consent withdrawal, please see our Privacy Policy .

Reviewer #2: No

Reviewer #3: No

---

## [Author Response · Author response to Decision Letter 2]

11 Apr 2025

Editor Comments

The structure also should be enhanced: there are separate analysis on which the rationale and the rationale of the sequence is not clear. Also regarding whether to use a scale vs all dichotomous items. The methods section and the research objectives should be made clearer so that the analysis flows from these. There is also a problem on sample size that might not be adequate for some of the more sophisticated analysis. As you know, due to multicollinearity, variances in multiple variable methods are higher than for simple two-variable analysis that you have used to ascertain sample size. This should be acknowledged as a limitation.

Response: Thank you for pointing this out and recommendations. We revised the structure of the ‘data analysis procedure’ section. You can see the changes. We acknowledge that a Likert scale could have provided more nuanced insights into the degree of influence. We have also addressed this concern in the limitation section. However, we choose a dichotomous scale to capture a clear distinction between individuals who perceive environmental concerns as influencing their childbearing decisions and those who do not. This approach simplifies analysis and eases interpretation in statistical modelling. In addition, this item did not show a high Cronbach’s Alpha value; therefore, analysing the items independently as dichotomous variables enables a nuanced understanding of each concern instead of using it as a combined score. This justification has also been included in the manuscript. Moreover, while our sample size is adequate for descriptive and bivariate analyses, it may be limited for more complex multivariate models. We acknowledge this as a limitation in the manuscript.

In addition, citation style is not according to PLOS ONE numbered style. This needs to be fixed.

Response: Thank you for your comment. We have corrected the citation style according to the PLOS ONE numbered style.

The questionnaire, currently included as an appendix to the main manuscript, should be provided as a separate file.

Response: Corrected

Regarding the first reviewer, who was unavailable, apparently the points have been addressed. The second reviewer provides some optional suggestions. Reviewer 3 indicates that language must be improved. The editor concurs on the need to copy-edit the manuscript.

Examples of issues that should be addressed, but there are many. The whole writing up needs to be improved.

• L. 83: The reading is now very different. “Highly productive” does not seem to apply given the location of Bangladesh in GDP rankings. The beautifulness of the rivers is also a subjective term such as those reviewer 1 was indicating in the previous revision.

Response: Thank you for the comment. The confusion has been cleared.

• L. 196-198: Remove consideration on the buildings of SUST unless it is deemed relevant for some reason. Discussion should be based on academic program and level.

Response: Sentences have been removed.

• L. 245: literature should be singular.

Response: Corrected

• Example of needed rephrasing is “measurement and analysis”. Some issues with language: “The researchers coded the following predictors into

Response: Measurement and analysis rephrased into ‘Data analysis procedure’. The statement regarding “The researchers coded the following predictors into” has been removed as it is not well-fitted in the manuscript.

• 269 categorical variables for binary logistic regression and other analyses:”. Why mixing the analysis with the coding? Was the coding ex-post carried out by the researchers as this sentence implies? It seems this was a closed questionnaire so that you should not mention “coding” at all. More important: we do not need to know the numbers of the codes at all! Also, on this: “which faculty 272 members are currently studying” you mean “field of study”. As commented, grammar needs to be revised and improved.

Response: The confusion has been cleared and modified according to your suggestion.

• L. 278: What is meant by “graphical distributions were created using RStudio”. You mean graphics were produced with R (RStudio is the gui. “Graphical distributions” is meaningless. Statistical distributions are not graphical.)

Response: Corrected.

• Figure 3: The axis should be clearly identified: percentage what?

Response: Corrected all of the figures.

• Table 5 should include the respective Ns as a new column.

Response: Corrected.

• Table 6 is missing N

Response: Corrected.

• L. 467 and later: It is not specified whether the poisson regression includes also those reporting not wanting to have children codified as 0 children. It should, otherwise you are not dealing with the complete distribution.

Response: The Poisson regression model was conducted only among respondents who reported intending to have children, as the follow-up question on number of desired children was asked conditionally (You can see the questionnaire and Table 4). As such, individuals who stated they did not intend to have children were not included in the model.

• For discussion: You are including as a variable “Vulnerability to home locality” (which probably should be “vulnerability of”. Note this is student’s perception. It might well be that students living in the same locality fill this up differently.

Response: Modified and rephrased into perceived vulnerability of home locality to climate change or extreme weather events.

Review Comments to the Author

Reviewer #2: Context of Study: While the paper provides an environmental description of Bangladesh, it would be beneficial to include a more detailed demographic context, particularly characteristics relevant to the studied participants (e.g., religious beliefs, education levels, socioeconomic status). Additionally, comparing these demographics to national data would help illustrate the representativeness of the sample and strengthen the study’s generalizability.

Response: Thank you for your comment. We have included national data to represent the sample. We have also compared the national data in our findings. You can see the changes in the ‘Context of the Study’ and ‘Result-Descriptive Statistics’ sections.

Limitations: Has any effort been made to ensure that respondents interpreted the questions consistently? While it is commendable that the study acknowledges its limitations, it would be helpful to outline any steps taken to minimize potential misunderstandings, such as piloting the survey, providing clarifications, or conducting follow-up checks. This would enhance the study’s methodological rigor.

Response: Thank you for your suggestion. To ensure the clarity and consistency of the questionnaire, a pilot test was conducted with 12 participants (you can see this data collection techniques section). You can also see the changes in the limitation section.

Reviewer #3: I appreciate the work you have put in to respond to all three reviewers, and I think your manuscript is a lot better now. My two main concerns now are language and the use of Sasser (2024).

- Language: I find parts of the manuscript to be a bit hard to read, as many sentences are long and sometimes confusing. You should go through the whole manuscript trying to make the language clearer. Example: "The analysis revealed that factors such as gender, religion, CGPA, disciplinary backgrounds, home locality susceptible to climate change or EWEs, childlessness have a beneficial effect on the environment, and going childless is a better way to help the environment than recycling, found to be statistically significant." (p. 30, lines 536-540). Furthermore, the last two factors listed here seem more like items to me?

Response: Thank you for the suggestion. The sentence has been modified. We have also improved the language with clarity.

- Sasser (2024): You write that you have tried to "incorporate relevant discussions on climate justice and reproductive justice related to our findings". The way I read your discussion, you refer briefly to Sasser (2024) without actually discussing how your findings relate to her arguments. This must be fixed.

Response: Modified. You can see the arguments in the discussion section.

---

## [Decision Letter · Decision Letter 2]

24 Apr 2025

PONE-D-24-46311R2The nexus between environmental concern and future childbearing aspirations among university students in BangladeshPLOS ONE

Dear Dr. Atiqul Haq,

Thank you for submitting your manuscript to PLOS ONE. After careful consideration, we feel that it has merit but does not fully meet PLOS ONE’s publication criteria as it currently stands. Therefore, we invite you to submit a revised version of the manuscript that addresses the points raised during the review process.

A reviewer is still concerned about interpretation issues that you should fix.In addition, there are still opinionated terms that should be avoided. One instance is the expression "adverse reproductive consequences" that appears twice. You should replace it by a non-opinionated term such as lower reproductive intention or something along this line. In general the complete sentence " Our research also indicates that environmental degradation 645 can result in population decline due to adverse reproductive consequences. " shoud be removed since this is clearly beyond the analysis of the paper.

We look forward to receiving your revised manuscript.

Kind regards,

José Antonio Ortega, Ph.D.

Academic Editor

PLOS ONE

Journal Requirements:

Reviewers' comments:

Reviewer's Responses to Questions

**Comments to the Author**

1. If the authors have adequately addressed your comments raised in a previous round of review and you feel that this manuscript is now acceptable for publication, you may indicate that here to bypass the “Comments to the Author” section, enter your conflict of interest statement in the “Confidential to Editor” section, and submit your "Accept" recommendation.

Reviewer #3: (No Response)

2. Is the manuscript technically sound, and do the data support the conclusions?

Reviewer #3: Yes

3. Has the statistical analysis been performed appropriately and rigorously? 

Reviewer #3: I Don't Know

4. Have the authors made all data underlying the findings in their manuscript fully available?

Reviewer #3: Yes

5. Is the manuscript presented in an intelligible fashion and written in standard English?

Reviewer #3: No

6. Review Comments to the Author

Reviewer #3: I appreciate that you have elaborated on how your work might relate to Sasser's. It will probably seem to you that I have a hang-up on her work, but since I now have commented on this aspect of your manuscript, I think I am obliged to follow up on it. The way you discuss your results in relation to Sasser, is still not convincing to me. Frankly, I find it a bit hard to follow what your argument actually is, other than that you both address reproductive issues and environmental concerns. I read your discussion as if you are saying that it makes less sense to view reproductive issues as a product of individual concern, when the current context is one of environmental degradation (i.e. that the context of environmental degradation is more relevant than individuals' concerns when it comes to reproductive choices). If this is what you are saying, this should be spelled out more clearly. If you, on the other hand, argue that your engagement with Sasser is mostly through that you both emphasize climate anxieties, this should also be made clearer. If there is a combination, as suggested by the sentence "While the result reflects the environmental concern, it also raises critical questions about whether such beliefs are freely chosen or shaped by structural factors", then this must also be written out as clearer arguments based on your results and how they contribute to Sasser's framework.

My specific concern with how you discuss Sasser, mirrors a more general concern regarding the clarity of your discussion. For instance, when you have revised the sentences on bottom p. 31 and top p. 32, I think you could still do more to make the language clear, correct, and unambiguous. To illustrate what I mean, I'll provide specific suggestions for these two sentences:

Current: "The analysis revealed that factors such as gender, religion, CGPA, disciplinary backgrounds, perceived vulnerability of home locality to climate change or EWEs were significantly associated with future childbearing intention."

Suggestion: Add an "and" before "perceived".

Current: "In addition, two specific items- childlessness has a beneficial effect on the environment, and going childless is a better way to help the environment than recycling- were also found to be statistically significant."

Suggestion:"In addition, two specific items (i.e. "childlessness has a beneficial effect on the environment"; "going childless is a better way to help the environment than recycling") were also found to be statistically significant."

7. PLOS authors have the option to publish the peer review history of their article (what does this mean? ). If published, this will include your full peer review and any attached files.

**Do you want your identity to be public for this peer review?** For information about this choice, including consent withdrawal, please see our Privacy Policy .

Reviewer #3: No

---

## [Author Response · Author response to Decision Letter 3]

1 May 2025

Editor Comment:

In addition, there are still opinionated terms that should be avoided. One instance is the expression "adverse reproductive consequences" that appears twice. You should replace it by a non-opinionated term such as lower reproductive intention or something along this line. In general the complete sentence " Our research also indicates that environmental degradation 645 can result in population decline due to adverse reproductive consequences. " shoud be removed since this is clearly beyond the analysis of the paper.

Response: Thank you for your suggestion. We removed the statement from this draft.

Journal Requirements:

Response: No retracted articles included in the manuscript.

Reviewer #3: I appreciate that you have elaborated on how your work might relate to Sasser's. It will probably seem to you that I have a hang-up on her work, but since I now have commented on this aspect of your manuscript, I think I am obliged to follow up on it. The way you discuss your results in relation to Sasser, is still not convincing to me. Frankly, I find it a bit hard to follow what your argument actually is, other than that you both address reproductive issues and environmental concerns. I read your discussion as if you are saying that it makes less sense to view reproductive issues as a product of individual concern, when the current context is one of environmental degradation (i.e. that the context of environmental degradation is more relevant than individuals' concerns when it comes to reproductive choices). If this is what you are saying, this should be spelled out more clearly. If you, on the other hand, argue that your engagement with Sasser is mostly through that you both emphasize climate anxieties, this should also be made clearer. If there is a combination, as suggested by the sentence "While the result reflects the environmental concern, it also raises critical questions about whether such beliefs are freely chosen or shaped by structural factors", then this must also be written out as clearer arguments based on your results and how they contribute to Sasser's framework.

Response: Thank you for your thoughtful follow-up comment. We appreciate your continued attention to how our paper relates to Sasser’s ideas. We now see that we could have explained more clearly how our findings connect with her argument. In the revised version of the paper, we explain it with clarity. We hope this more explicit engagement will address your concern and clarify our position in relation to Sasser’s work. You can see the changes in discussion section.

My specific concern with how you discuss Sasser, mirrors a more general concern regarding the clarity of your discussion. For instance, when you have revised the sentences on bottom p. 31 and top p. 32, I think you could still do more to make the language clear, correct, and unambiguous. To illustrate what I mean, I'll provide specific suggestions for these two sentences:

Current: "The analysis revealed that factors such as gender, religion, CGPA, disciplinary backgrounds, perceived vulnerability of home locality to climate change or EWEs were significantly associated with future childbearing intention."

Suggestion: Add an "and" before "perceived".

Response: Thank you for your comment. You can see the changes

Current: "In addition, two specific items- childlessness has a beneficial effect on the environment, and going childless is a better way to help the environment than recycling- were also found to be statistically significant."

Suggestion: "In addition, two specific items (i.e. "childlessness has a beneficial effect on the environment"; "going childless is a better way to help the environment than recycling") were also found to be statistically significant."

Response: Corrected.

---

## [Editor Report · Decision Letter 3]

9 May 2025

PONE-D-24-46311R3The nexus between environmental concern and future childbearing aspirations among university students in BangladeshPLOS ONE

Dear Dr. Atiqul Haq,

Thank you for submitting your manuscript to PLOS ONE. After careful consideration, we feel that it has merit but does not fully meet PLOS ONE’s publication criteria as it currently stands. Therefore, we invite you to submit a revised version of the manuscript that addresses the points raised during the review process.

**The reviewer that was still pending acceptance was not available. The comments seem to be implemented.**
**My specific comments on avoiding opinionated terms has been implemented, but I am missing among the limitations of the study an acknowledgement that this is an observational study and that no causal relation between specific concerns and childbearing intentions can be inferred. There could be non-causal mechanisms such as people with certain traits having correlated opinions and attitudes in the dimensions explored. Please introduce a paragraph along these lines in the limitations, and the paper could be published.**

We look forward to receiving your revised manuscript.

Kind regards,

José Antonio Ortega, Ph.D.

Academic Editor

PLOS ONE
---

## [Author Response · Author response to Decision Letter 4]

10 May 2025

Editor Comment:

My specific comments on avoiding opinionated terms has been implemented, but I am missing among the limitations of the study an acknowledgement that this is an observational study and that no causal relation between specific concerns and childbearing intentions can be inferred. There could be non-causal mechanisms such as people with certain traits having correlated opinions and attitudes in the dimensions explored. Please introduce a paragraph along these lines in the limitations, and the paper could be published.

Response: Thank you for your suggestion. We included your suggested lines in the limitations section. We hope that you will find our revised version publishable now.

---

## [Editor Report · Decision Letter 4]

13 May 2025

The nexus between environmental concern and future childbearing aspirations among university students in Bangladesh

PONE-D-24-46311R4

Dear Dr. Atiqul Haq,

We’re pleased to inform you that your manuscript has been judged scientifically suitable for publication and will be formally accepted for publication once it meets all outstanding technical requirements.

Kind regards,

José Antonio Ortega, Ph.D.

Academic Editor

PLOS ONE

Additional Editor Comments (optional):

The limitation has been added as suggested on the impossibility of inferring causal claims from observational data. The reviewers' concerns had been addressed in previous drafts.
---

## [Editor Report · Acceptance letter]

PONE-D-24-46311R4

PLOS ONE

Dear Dr. Atiqul Haq,

I'm pleased to inform you that your manuscript has been deemed suitable for publication in PLOS ONE. Congratulations! Your manuscript is now being handed over to our production team.

Kind regards,

on behalf of

Dr. José Antonio Ortega

Academic Editor

PLOS ONE